# Novel Injectable Fluorescent Polymeric Nanocarriers for Intervertebral Disc Application

**DOI:** 10.3390/jfb14020052

**Published:** 2023-01-17

**Authors:** Michael R. Arul, Changli Zhang, Ibtihal Alahmadi, Isaac L. Moss, Yeshavanth Kumar Banasavadi-Siddegowda, Sama Abdulmalik, Svenja Illien-Junger, Sangamesh G. Kumbar

**Affiliations:** 1Department of Orthopedic Surgery, University of Connecticut Health, Farmington, CT 06030, USA; 2Department of Orthopedic Surgery, Emory University, Atlanta, GA 30308, USA; 3Department of Biomedical Engineering, University of Connecticut, Storrs, CT 06269, USA; 4Surgical Neurology Branch, National Institute of Neurological Disorders and Stroke, National Institutes of Health, Bethesda, MD 20892, USA; 5Department of Materials Science and Engineering, University of Connecticut, Storrs, CT 06269, USA

**Keywords:** fluorescent nanoparticles, drug delivery, injectable, intervertebral disc, cellulose, polycaprolactone

## Abstract

Damage to intervertebral discs (IVD) can lead to chronic pain and disability, and no current treatments can fully restore their function. Some non-surgical treatments have shown promise; however, these approaches are generally limited by burst release and poor localization of diverse molecules. In this proof-of-concept study, we developed a nanoparticle (NP) delivery system to efficiently deliver high- and low-solubility drug molecules. Nanoparticles of cellulose acetate and polycaprolactone-polyethylene glycol conjugated with 1-oxo-1H-pyrido [2,1-b][1,3]benzoxazole-3-carboxylic acid (PBC), a novel fluorescent dye, were prepared by the oil-in-water emulsion. Two drugs, a water insoluble indomethacin (IND) and a water soluble 4-aminopyridine (4-AP), were used to study their release patterns. Electron microscopy confirmed the spherical nature and rough surface of nanoparticles. The particle size analysis revealed a hydrodynamic radius ranging ~150–162 nm based on dynamic light scattering. Zeta potential increased with PBC conjugation implying their enhanced stability. IND encapsulation efficiency was almost 3-fold higher than 4-AP, with release lasting up to 4 days, signifying enhanced solubility, while the release of 4-AP continued for up to 7 days. Nanoparticles and their drug formulations did not show any apparent cytotoxicity and were taken up by human IVD nucleus pulposus cells. When injected into coccygeal mouse IVDs in vivo, the nanoparticles remained within the nucleus pulposus cells and the injection site of the nucleus pulposus and annulus fibrosus of the IVD. These fluorescent nano-formulations may serve as a platform technology to deliver therapeutic agents to IVDs and other tissues that require localized drug injections.

## 1. Introduction

Every year in the US, approximately 5.7 million people suffer from intervertebral disc (IVD)-related diseases that cause back pain and disability [1,2]. The degenerative disc disease market size is expected to grow by USD 1.03 billion from 2021 to 2026. Today’s non-operative and operative treatments for IVD degeneration mostly focus on temporary pain relief but fail to prevent IVD degeneration progression. This has led to overreliance on oral, systemic pain medications, which has contributed to the current opioid crisis [3,4].

IVD degeneration is a pathological process characterized by extracellular matrix (ECM) degeneration, chronic inflammation, and increased apoptosis [5]. The IVD is a fibrocartilaginous tissue that lies between two adjacent vertebral bodies and acts as a shock absorber [6]. It consists of a central gelatinous nucleus pulposus (NP), surrounded by annulus fibrosus (AF), and a cartilaginous endplate anchoring the IVD to the vertebrae [7]. The causes for IVD degeneration are multifactorial and include a sedative lifestyle, trauma-induced degeneration, occupation (continuously altered loading of the IVD), aging, and heritage [8]. IVDs have limited healing potential, and the ideal treatment for IVD degeneration would restore the ECM to its healthy state, which includes healthy NP cells and an intact ECM metabolism.

Biological approaches including gene therapy and stem cell-based therapy are emerging as new avenues for IVD degeneration treatment, but both have their limitations, such as low transfection efficiency, vector, cell leakage, and cell differentiation [9,10,11,12]. Currently, injections of bioactive agents or growth factors, such as bone morphogenetic protein-7 [13], the TNF-α signaling inhibitor fexofenadine [14], and a small-molecule inhibitor of the Wnt pathway [15], have been shown to promote proteoglycan content, improve IVD structural integrity and disc height, or reduce inflammation, all of which contribute to IVD regeneration. The most common problems with many bioactive agents include poor aqueous solubility resulting in poor bioavailability [16,17,18].

Several hydrogels and microparticle depot formulations have been explored for the delivery of therapeutics to IVDs and are successful in improving the drug residence time [19]. Though these depot formulations avoid repeated drug injections, they continue to face challenges specific to carrier systems [20]. Hydrogel formulations are limited to the delivery of hydrophobic drugs and often release the drug in a very short time due to high water content [21]. Additionally, injected hydrogels can occupy IVD space for an extended period, and because of their relatively large volume and slow degradation rate in combination with the avascular nature of IVDs, their byproduct clearance is limited, resulting in prolonged accumulation of degradation products within the tissue [22]. Microparticle formulations allow the sustained release of both hydrophobic and hydrophilic drugs; however, size may be a concern in their applicability [23]. Nanotechnology and nanocarriers have revolutionized the medical field by providing novel tools for imaging, diagnosis, or the delivery of drugs to treat various diseases [24,25] Nanoformulations such as liposomes and micelles offer several advantages over hydrogels and microparticles as injectables that enhance drug solubility, stability, injectability, cell internalization, targeting, and degradation [26]. Polymeric nanoparticles additionally offer enhanced aqueous stability over micelles and liposomes and have been proven beneficial in preclinical models [24,25].

Here, we synthesized and characterized injectable, fluorescent polymeric nanoparticles using natural and synthetic polymer backbones. We tested their efficacy in the sustained release of the surrogate drug indomethacin (IND), which is hydrophobic, and the freely water-soluble 4-aminopyridine (4-AP). The chosen drugs resemble the majority of the small drug molecules that are used to prevent IVD degeneration.

We developed an efficient synthetic route for a novel citric acid-based small fluorescent molecule with an emission and absorbance wavelength in the range of 450–550 nm and 350–450 nm based on the functional group conjugation. We used cellulose acetate (CA) and a copolymer polycaprolactone-polyethylene glycol (PCL-PEG) as polymer backbones to conjugate the fluorescent dye. These polymers were processed into nanoparticles by a standard oil-in-water emulsion system. Injectable nanoparticles with two model drugs, namely, IND and 4-AP, were further characterized for chemical structure, property, In vitro drug release, and cytotoxicity. As a proof of concept, nanoparticles were tested for their injectability into mice coccygeal IVDs and subsequent fluorescent imaging to confirm localization.

## 2. Material and Methods

### 2.1. Chemical Reagents

Cellulose acetate (MW: 50,000) (CA), caprolactone (CL), polyethylene glycol (PEG, MW 3500), 2-amino phenol, N, N′-dicyclohexylcarbodiimide (DCC), polyvinyl alcohol (30,000–70,000), stannous octoate (Sn(Oct)_2_), and 4-dimethyl aminopyridine (DMAP) were purchased from Sigma-Aldrich (St. Louis, MO, USA). 4-Aminopyridine (>99%) was purchased from Alomone Labs (Jerusalem, Israel). Dimethylformamide (DMF), citric acid, phosphoric acid, regenerated cellulose dialysis bag (MWCO of 3500 Da), Human Dermal Fibroblasts, adult (HDFa) (Cascade Biologics™), cell culture media DMEM, fetal bovine serum (FBS), phosphate-buffered saline (PBS), and LIVE/DEAD™ viability/cytotoxicity kit were purchased from Fisher Scientific (Fair Lawn, NJ, USA). CellTiter 96^®^ Aqueous One Solution Cell Proliferation Assay (MTS) was obtained from Promega (Madison, WI, USA). Ultrapure pure water (Millipore) and HPLC-grade solvents were used without further purification.

### 2.2. Synthesis of 1-Oxo-1H-pyrido[2,1-b][1,3]benzoxazole-3-carboxylic Acid (PBC)

The fluorescent dye 1-oxo-1H-pyrido[2,1-b][1,3]benzoxazole-3-carboxylic (PBC) acid was synthesized by reacting a 3:1 molar ratio of 2-amino phenol and citric acid in concentrated orthophosphoric acid as a solvent [27]. In brief, a round bottom (RB) flask fitted with a water condenser was charged with reaction reagents refluxed at 180 °C for 24 h and precipitated using Milli-Q water. The reaction mixture was washed repeatedly to remove water-soluble reactants and crystallized in DMF to obtain pure PBC for further characterization.

### 2.3. Synthesis of Polycaprolactone-Co-Polyethylene (PCL-PEG) Copolymer

The PCL-PEG copolymer synthesis was carried out by CL ring-opening polymerization in the presence of Sn(Oct)_2_ catalyst. Five mg of PEG (Mw 3500) was dried in an RB flask by dissolving in toluene and removing the solvent with a rotary evaporator (Buchi rotavapor r-100). The bath temperature was maintained at 40 °C with an atm pressure of 77. To PEG in an RB flask, 10 g of dried CL was added using a syringe through the rubber septum under a nitrogen atmosphere to avoid exposure to moisture. Following stirring for an hour, 20 µL of Sn(Oct)_2_ was added to the reaction mixture using a syringe. The reaction temperature was maintained at 120 °C using an oil bath with a constant nitrogen gas purging for 7 h. The viscous reaction mixture was cooled, dissolved in 50 mL dichloromethane, and precipitated in a cold 200 mL of methanol/hexane (75:25 ratio) solvent mixture. The precipitate was redissolved in dichloromethane and re-precipitated in methanol/hexane to remove impurities and the copolymer was dried at 50 °C under a vacuum for two days.

### 2.4. Synthesis of PBC Dye-Conjugated CA and PCL-PEG Polymers

1g of CA and 0.95 g of PBC representing 4.19 millimoles equimolar ratio were dissolved in 50 mL anhydrous DMF in an RB flask while purging nitrogen gas. Measures of 0.15 g 4-DMAP (1.24 mM) and 1.53 g DCC (7.45 mM) in 5.0 mL anhydrous DMF were added to the RB flask. The reaction mixture temperature was kept at 25 °C for the first hour and raised to 100 °C for an additional 24 h. The reaction was cooled and filtered using a Whatman filter 0.45 µm to remove di-cyclohexyl urea, and the excess solvent was removed using a rotary evaporator. The thick viscous solution was precipitated in cold water and the precipitate was washed repeatedly with Milli-Q water. These dye-conjugated polymers were redissolved in DMF and precipitated in water to remove impurities by repeating this process twice. The final brown product was vacuum dried at 60 °C and kept desiccated until further use. The synthetic and purification protocol for polycaprolactone-co-polyethylene conjugated dye (PCL-PEG-PBC) remained identical.

### 2.5. Fabrication of Nanoparticles and Drug-Loading

Nanoparticles were fabricated following an oil-in-water emulsion and solvent evaporation technique [28]. In brief, 1 g of CA-PBC and 100 mg of IND were dissolved in 20 mL of dichloromethane. This solution was emulsified into 35 mL of 2 wt. % polyvinyl alcohol (PVA) aqueous solution via sonication. The sonication was carried out using a TU-150E4 probe sonicator for five minutes with a 60 s time interval between the cycle at 100 W. The emulsion was stirred for an additional 4 h to evaporate the solvent. The nanoparticles were isolated by centrifuging at 17709 rcf for 20 min at 4 °C. Nanoparticles were washed repeatedly to remove residual PVA and surface-attached drug crystals. Drug-loaded nanoparticles were freeze-dried using a lyophilizer and kept desiccated until further use. Based on our prior work a total of two drug loading concentrations (5 and 10 wt% of the polymer) were considered based on their high encapsulation efficiencies [29]. Similar procedures were followed to incorporate 4-AP and other nanocarriers with and without the drug for CA, PCL-PEG, and PCL-PEG-PBC systems.

### 2.6. Characterization of Polymeric Nanoparticles

FTIR spectroscopy was used to identify functional groups in the newly synthesized polymers. The Nicolet OMNIC^®^ software was used to record the spectra in 400–4000 cm^−1^. Each spectrum is an average of 320 scans. An 800 MHz NMR spectrometer (Varian) was used to record ^1^H NMR to confirm structures for PBC, CA, PCL-PEG, CA-PBC, and PCL-PEG-PBC. Spectra were recorded in DMF and CDCl_3_ with 0.03% (*v*/*v*) tetramethylsilane as an internal standard. Powder XRD studies were carried out to characterize the drug crystallinity and its distribution within nanoparticles. The diffraction patterns were recorded on a Bruker D2 Phaser equipped with Cu-Kα (λ 1.54056) radiation at 40 kV and 30 mA (Bruker AXS, Madison, WI, USA). Nano formulations of CA, CA-PBC, PCL-PEG, PCL-PEG-PBC, and PBC with and without drugs were recorded from 58 to 608 (2u) at a scanning speed of 0.2 deg/s. The surface morphology of nanoparticles was captured using an SEM (JEOL JSM-6335 F, JEOL USA, Inc., Peabody, MA, USA). Nanoformulations were coated with Au/Pd using a Polaron E5100 sputtering system (Quorum Technologies, East Sussex, UK) before imaging. The duration of sputter coating was 1 min. Dynamic light scattering (DLS) was employed to assess the particle size of nanoparticles. Nanoformulations at a 0.1 wt% concentration were dispersed in Milli-Q water at 25 °C, and a 632.6 nm laser beam instrument was used to record the scattering pattern (ALV/CGS-8F/4 (ALV compact goniometer system, Langen, Germany). The zeta potential for each nanoformulation was measured in PBS suspensions at 25 °C on a Zetasizer Nano ZS (Malvern Instruments Ltd., Malvern, UK). These results are an average of 10 different measurements of each formulation (*n* = 5).

### 2.7. Determination of Drug Encapsulation Efficiency

The encapsulated drug in nanoparticles was extracted in ethanol using a probe sonicator (TU-150E4, Xian Toption Instrument Co., Ltd., Xi’an, China) for quantification. In brief, 5 mg of each drug formulation was dispersed in 5 mL of ethanol in a glass vial and sonicated for 10 min with a 60 s time interval between the cycle at 100 W. This solution was centrifuged at 7871 rcf for 10 min. The supernatant was subjected to drug content analysis using a UV spectrophotometer (Genesys 10 S, Thermo Fisher Scientific, Waltham, MA, USA). The absorption for IND was measured at 320 nm and 4-AP at 262 nm. The absorption value was converted into concentration using the standard curve for each drug. The % encapsulation efficiency (*%EE*) was calculated using Equation (1). For each formulation, a sample size of *n* = 3 was used and reported as the Avg ± Std. Dev.
(1)$$\%EE=\frac{Actual\;Drug\;Concn.}{\;Drug\;Concn.\;Used\;in\;Formulation}\times 100$$

### 2.8. In Vitro Drug Release

The drug release patterns for all nanoparticle formulations were carried out in PBS pH 7.4 at 37 °C using an orbital shaker. In brief, nanoformulations weighing ~100 ± 5 mg were placed in dialysis bags (MWCO 12,000 Da) with 1 mL PBS, tied, and incubated in 20 mL of PBS in 50 mL Eppendorf tubes were used to maintain a sink condition. 1 mL of the dissolution media was collected after 0.5, 1, 3, and 6 h, and thereafter daily, replenished with fresh PBS, and quantified using a UV-Vis spectrophotometer with a standard curve at 320 or 262 nm for IND or 4-AP, respectively. For each formulation, a sample size of *n* = 3 was used and reported as the Avg ± Std. Dev.

### 2.9. Drug Transport Studies

The drug diffusion parameters nanoformulations were analyzed using Higuchi (Equation (2)) and Korsmeyer–Peppas (Equation (3)) models. The Higuchi model describes the release of drugs from an insoluble matrix as a square root of a time-dependent process based on the Fickian diffusion Equation. The data obtained were plotted as cumulative percentage drug release versus square root of time. The Korsmeyer–Peppas model shows the drug release from a polymeric system through release mechanisms such as the diffusion of water into the matrix, swelling of the matrix, and dissolution of the matrix, where *M_t_*/*M*_0_ or *M_t_*/*M_∞_* are fractional drug releases at time *t* (in hours); *k* is the characteristic constant for a drug-polymer system; *n*, the diffusion exponent, suggests the release mechanism.
(2)$$\frac{M_{t}}{M_{0}}=Kt^{\frac{1}{2}}$$

A value of *n* = 0.5 indicates Fickian transport, while *n* = 1 indicates Case II transport. The intermediary values ranging between 0.5 and 1.0 are indicative of anomalous transport.
(3)$$\frac{M_{t}}{M_{\infty }}=kt^{n}$$

### 2.10. In Vitro Fibroblast Nanoparticle Uptake and Viability

In vitro nanoparticle uptake and its visualization were carried out using Human Dermal Fibroblasts, adult (HDFa; Cascade Biologics™). Passage 7 was used for all cell culture experiments. In brief, ~5000 cells/well were seeded in a 48-well plate, allowed to attach, and cultured in high glucose DMEM (no glutamine, no phenol red with 10% FBS (Fetal bovine serum), 1% Glutamax supplement, 1% Sodium pyruvate, and 1% Penicillin) (supplemented with the drug nanocarriers). Toxicity studies looked at free drug serial dilutions and found them to be nontoxic at 10 µg/mL for 4-AP and 50 µg/mL for IND. A typical 12mL of culture media contained 50–150 mg of CA, PCL-PEG, and PBC-conjugated nanoparticles with 4-AP and IND drugs to represent the identified nontoxic two-drug concentrations. Cells in 48 well-plates were treated with 300 µL of nanoparticle-containing media. Therefore, nanoparticles with 4-AP (10 µg/mL) or IND (50 µg/mL) were prepared in DMEM medium (*n* = 5) for cell culture experiments. These cultures were imaged post 1 and 3 days of culture. Cells were stained with the LIVE/DEAD™ Viability/Cytotoxicity Kit as per the manufacturer’s instructions (Thermofisher Scientific Catalog number: L3224). Cells were imaged using a Nikon Eclipse E600 fluorescence microscope (Nikon Instruments Inc, Melville, NY, USA) at 20× magnification. The FITC channel was used to image live cells and TEXAS red channel was used for the dead cells and the DAPI channel was used to image nanoparticles.

### 2.11. Nanoparticle Formulation Effect on Cell Proliferation

The effect of nanoformulations on fibroblast proliferation over 72 h was estimated using CellTiter 96^®^ AQueous One Solution Cell Proliferation Assay (MTS). In brief, ~5000/well was seeded in a 48-well plate, allowed to attach, and DMEM media supplemented with the calculated amount of drug nanocarriers were added. Cells were treated with two different loadings of 4-AP (10 µg/mL) and IND (50 µg/mL) prepared using different polymeric carriers including CA, CA-PBC, PCL-PEG, and PCL-PEG-PBC nanoparticles. Controls included untreated cells (untreated control) and cells treated with free drugs at identical concentrations as nanocarrier (drug control). Cell cultures at 24 h and 72 h treatment were subjected to MTS assay by adding the CellTiter reagent (20 µL/well), followed by incubation at 37 °C for 1 h. A surfactant was added to quench the reaction and the absorbance of the media was measured at 490 nm on a plate reader (Biotek, Winooski, VT). For each formulation, a sample size of *n* = 5 was used and reported as the Avg ± Std. Dev.

### 2.12. Human NP Cell Isolation and Culture

Degenerated human lumbar NP cells (*n* = 2; Pfirrmann grade 4 and 5) were harvested from patients undergoing discectomy surgery following a protocol approved by the Institutional Review Board at the Emory University School of Medicine. NP cells were isolated from NP tissue that could be clearly distinguished from AF tissue and did not contain unidentifiable tissues or blood vessels. NP tissues were washed in 70% ethanol, washing solution (1xPBS, 3% pen/strep, 1.5% Amphotericin B), and 1xPBS. Tissues were then diced into small pieces (~1 mm^3^) and cells were released by digestion in 0.2% protease (Sigma Aldrich, P5147-1G) for 1 h, followed by 0.025% collagenase P (Roche Diagnostics, 40341623) digestion for 4 h. Isolated NP cells were filtered through a 100 μm cell strainer and rinsed twice in 1xPBS. NP cells were expanded in low glucose DMEM media (1% glucose + 10% FBS + 1% Penicillin–Streptomycin) under normoxia (21% O_2_) and 5% CO_2_, at 37 °C in a humidified atmosphere. The culture medium was changed every other day. Cells of passages 2–3 were used for all experiments.

### 2.13. Nanoparticle Uptake in Human NP Cells

Because degenerated human NP cells are quiescent and have a slow cell turnover, we first established whether nanoparticles could be taken up by degenerated primary NP cells. NP cells (*n* = 2, biological replicates, passage 2–3) were seeded in 35 mm glass-bottom cell culture dishes at a density of 4 × 10^5^ cells/mL. After NP cells reached 80% confluency, 0.015 μg/mL nanoparticles (concentration of the in vivo experiments) or 1 x PBS suspension were added to the culture media (low glucose DMEM: 1% glucose + 10% FBS + 1% Penicillin–Streptomycin) for 3 days. NP cells were washed with 1 x PBS 3 times and fixed with fresh 4% paraformaldehyde (PFA) for 15 min at room temperature. To visualize cell morphology, cells were stained with CellMask^TM^ Plasma Membrane Stains (ThermoFisher, C10046) for 10 min at 37 °C. Cells were then mounted with Histomount Mounting Solution (ThermoFisher, 00803) and imaged using an Inverted Confocal Microscope (Leica Stellaris5 with DMi8 stand; Leica, Wetzlar, Germany).

### 2.14. Mouse Surgeries and Sample Harvest

Mice experiments were conducted as per the recommendations stated in the Guide for the Care and Use of Laboratory Animals of the National Institutes of Health (U.S. Department of Health, Education, and Welfare, NIH 78-23, 1996) and approved by the Atlanta Veteran’s Affairs Medical Center’s Institutional Animal Care and Use Committee (Protocol V016-19). As a proof of concept to study the feasibility of nanoparticle injection, 2 tdTomato reporter mice (5-month-old mT/mG mice (JAX lab, strain # 007676) with C57BL/6J congenic background) were used for nanoparticle injections into coccygeal IVDs. Nanoparticle injections were performed under general anesthesia (2% isoflurane in oxygen) and sterile conditions. Coccygeal IVDs (cc4/5–cc8/10) were identified by palpation and exposed via a 2–4 mm dorsolateral incision. Nanoparticle suspension in PBS (0.015 μg/mL) and 1 μL was injected into healthy coccygeal IVDs. Healthy control IVDs were injected with 1 μL PBS (mouse 1 tail IVDs: nanoparticle injection *n* = 2, PBS injection *n* = 2; mouse 2: nanoparticle injection *n* = 4, PBS injection *n* = 2) using a 33-gauge needle attached to a Hamilton syringe (#7641-01, Hamilton, Reno, NV, USA). Incisions were sutured (Prolene 8-0 sutures; Ethicon, Somerville, NJ, USA) and mice were closely monitored to ensure the absence of intraoperative complications and allowed free activity in their cages with ad libitum access to food and water. Because the scope of this study was to test the feasibility of nanoparticle injection, rather than exploring the efficacy of drug treatments, mice were killed on day 1 after surgery via carbon dioxide inhalation. Tails were harvested and fixed in Z-fix. IVD motion segments were dissected using a Stereoscope (Leica). IVDs were cut mid-sagittal (using a #11 Scalpel blade), placed on glass-bottom Petri dishes with the mid-sagittal plane facing down, and mounted with 2% agarose.

### 2.15. Nanoparticle Uptake In Vivo

Nanoparticle uptake and localization 1 day after surgery were determined by fluorescence imaging using confocal microscopy (Leica Stellaris5 with DMi8 stand; Leica, Germany). Transgenic tdTomato reporter mice were used because their cells fluoresce (peak excitation at 554 nm) and could be used to visualize IVD morphology and cellularity. Nanoparticles have a peak excitation at ~431 nm and the DAPI channel was used to visualize the particles. Tiled images with z-stack (200 µm, step size 4 μm) were taken to localize nanoparticle accumulation within the entire IVD.

### 2.16. Statistical Analysis

All data are expressed as mean ± standard deviation (mean ± s.d.). The results were evaluated using a two-way analysis of variation (ANOVA) followed by multiple comparisons of the differences between groups with a confidence level of 95% (*p* < 0.05) using GraphPad Prism 8 (GraphPad Software, Inc. La Jolla, CA, USA).

## 3. Results

### 3.1. Synthesis of PBC, PCL-PEG, PBC-Conjugated PCL-PEG, and CA

Polymeric nanoparticles (NP) were synthesized to deliver high- and low-solubility drug molecules. Nanoparticles of CA and PCL-PEG conjugated with a fluorescent dye were prepared by the oil-in-water emulsion technique. The schematics for the synthesis of PBC dye, PCL-PEG, PBC dye-conjugated polymers CA, and PCL-PEG are presented in Figure 1. The aim of the PBC conjugation to the polymeric backbone was to modify matrix hydrophobicity and the ability to track fluorescent activity. These polymers were synthesized and purified for further characterization and application. Polymeric nanoparticles with and without drugs were prepared, freeze-dried, and kept desiccated until further use. The structures of the synthesized dye, polymers, and nanoformulations were confirmed by FTIR and NMR spectroscopy.

### 3.2. FTIR and ^1^H NMR Spectroscopy for PBC-Conjugated Polymers and Drug Formulations

The FTIR spectra for PBC, CA, CA-PBC, PCL-PEG, and PCL-PEG-PBC are presented in Appendix A. Table 1 depicts the different functional group wavelengths of PBC, CA, CA-PBC, PCL-PEG, and PCL-PEG-PBC. PBC dye has characteristic bands at 1722 cm^−1^ for -C=O stretching and amide at 1659 cm^−1^. The aromatic -CH_2_ bands of benzene at 2869 cm^−1^, -C-O stretching at 1097 cm^−1^, and free NH at 3138 cm^−1^ suggest the formation of the 2-pyridone ring from citric acid with 2-aminophenol. The FTIR spectra of PBC-conjugated CA showed −C=O stretching shift from 1738 to 1746 cm^−1^ and an amide peak from 1659 to 1668 cm^−1^ indicating the formation of a new ester linkage and conjugation. The shifting of hydroxyl group peaks of CA from 3479 to 3313 cm^−1^ indicates ester linkage formation.

The PCL-PEG FTIR spectra presented stretching vibrations for -C-O-C, -O-CH2-CH2, and -OH at 1733, 1107, and 3538 cm^−1^, respectively (Table 1). The PCL-PEG-PBC copolymer was confirmed by the formation of an ester bond between the PBC carboxylic group and the PCL-PEG hydroxyl group at 1735 and 3322 cm^−1^ [21].

The interactions between the functional groups for IND or 4-AP with polymeric nanoparticles were studied to identify possible drug matrix interactions (Table 1). The shifting of the distinct peaks of carbonyl (–C=O) stretching 1713 cm^−1^ of the drug to 1748, 1756, 1728, and 1724 cm^−1^ signifies potential interaction with the polymeric nanocarrier (please insert Arya et al.). Likewise, in the 4-AP formulations, the -NH_2_ stretching 1644 cm^−1^ slightly shifted to 1646, 1652, 1649, and 1651 cm^−1^.

Appendix A is the ^1^H NMR for PBC (DMF, δ/ppm) where -CH_2_ peaks of benzene ring are observed at δ 8.51, 7.80, 7.62, 7.81, and 7.55 ppm designated as c, d, e, and f. The peaks at 8.01 and 6.82 ppm designated as a and b belong to -CH- of the 6-membered ring, and it confirms the incorporation of citric acid into the 2-aminophenol compound. Appendix A is the ^1^H NMR of CA(CDCl_3_; δ/ppm were peaked at δ values of 1.51 ppm, 2.03 ppm, and 3.54 ppm identified as a, b, and c, belonging to the -CH backbone of a secondary alcohol, and -CH_3_ of ester methylene group of the secondary alcohol. The peaks d, e, and f with δ values at 3.37, 3.72, and 4.02 ppm are of the CA backbone. Appendix A is the ^1^H NMR of CA-PBC (CDCl_3_, δ/ppm) where the peak shifting was observed as compared to CA spectra. The peaks a, b, and c with δ values were shifted to 2.03, 1.65, and 3.57 ppm for CA-PBC as compared to CA. The shifting alcohol peak from 3.54 to 3.57 ppm confirms the formation of an ester bond between CA and PBC. The appearance of new peaks of the PBC benzene ring appeared at δ values of 6.61 ppm, 6.70, 7.42, and 8.63 ppm (g, h, i, k) confirming PBC conjugation to CA. The –CH (g’, j) groups of PBC peaks at δ values of 6.08 and 7.50 ppm further support CA-PBC conjugation.

The copolymer PCL-PEG ^1^H NMR spectra shown in Appendix A (DMF, δ/ppm) present characteristic CH_2_ protons in PCL repeat at 4.32 ppm, 1.45 ppm, 2.47 ppm, and 1.53 ppm (a’, d’, c, d). The -CH_2_ protons of PEG directly linked to the PCL are at 3.76 ppm (e) and the terminal -OH proton of PEG at 4.42 ppm (a). The -CH_2_ proton of the PEG at 3.64 ppm (b) and the -CH_2_OOC of PCL at 4.13 (a’) further confirm the formation of the copolymer. Appendix A is the ^1^H NMR spectra of PCL-PEG-PBC (DMF, δ/ppm) where the alcohol peak shifted from 3.54 ppm to 3.57 ppm, confirming the formation of an ester bond between PCL-PEG and PBC. The other characteristic peaks of PBC benzene (g, h, i, k) and -CH (g, j) of a 6-member ring confirm the formation of PCL-PEG-PBC [30].

### 3.3. SEM, Particle Size, Zeta Potential, and Absorbance Spectroscopy of Fluorescent Nanoparticles

The SEM images confirmed the spherical morphology with a rough surface (Figure 2a,b). The particle size DLS measurements showed the hydrodynamic radius varied between ~150-162 nm (Figure 2c). The particle size distribution with nanoparticle hydrodynamic radius is shown in Appendix A. The zeta potential of nanoparticles in PBS at pH 7.4 was −1.51, −4.42, −0.543, and −1.04 mV for CA, CA-PBC, PCL-PEG, and PCL-PEG-PBC, respectively (Figure 2d). The PBC-conjugated nanoparticles presented higher negative zeta potential confirming their enhanced stability as compared to placebo particles. The particle sizes for different nanoparticles included CA ~150–200 nm, CA-PBC ~170 nm, PCL-PEG~130–200 nm, and PCL-PEG-PBC ~125–200 nm (Appendix A). It is important to note that nanocarriers were spherical and presented a Gaussian distribution in their size. However, representative SEMs present the overall morphology, special nature, and size of larger size particles. The hydrodynamic radius measured by the dynamic light scattering method is a true representation of the overall particle size. The absorption wavelength for PBC and conjugated nanoparticles showed a broad peak that ranged from 350 to 450 nm and peaked at 425 nm (Appendix A).

### 3.4. XRD Analyses of Nanoparticles

The XRD patterns for the placebo polymeric nanoparticles and the drug formulations are presented in Figure 3. The PBC diffractogram showed distinct crystalline 2θ peaks at 11.4°, 15.2°, 19.1°, and 26.7° (Figure 3a). Upon conjugation to CA, these peaks disappeared, indicating the amorphous nature of both CA and CA-PBC nanocarriers. The PCL-PEG and PCL-PEG-PBC nanoparticles showed characteristic 2θ peaks at 21.1° and 23.7° indicating the crystalline nature of nanocarriers. Contributions of PBC dye toward the crystallinity of the nanocarriers are negligible due to the lower amount of the PBC on the polymer backbone at the molecular level. Sharp peaks in XRD diffractograms for IND and 4-AP nanoparticle formulations indicated that both drug molecules are highly crystalline (Figure 3b,c). However, in the nanoparticle formulations, the absence of crystalline domains due to the presence of the drug indicates molecular-level drug dispersion. The physical drug incorporation into the nanoparticles converted crystalline drugs into an amorphous form [31,32].

### 3.5. Drug Encapsulation Efficiency (%EE)

The %EE of IND varied from 70 to 85% based on the polymer carrier, while the %EE for 4-AP was very low and varied from 20 to 30% (Figure 4a,b). IND is an aprotic drug and has a low aqueous solubility, whereas 4-AP is freely water-soluble [33,34]. During the emulsion nanoparticle fabrication process, 4-AP is expected to dissolve more in water compared to IND, resulting in lower EE [35]. The standard curve for both IND and 4-AP is presented as Appendix A, respectively.

### 3.6. In Vitro Drug Release Profiles

The in vitro drug release profiles for IND and 4-AP in PBS pH 7.4 at 37 °C are presented in Figure 4. The release of IND from these nanoparticles continued for up to 4 days and the release rate was dependent on the carrier (Figure 4c). The release of IND lasted up to 3 days for CA carriers and up to 4 days for CA-PBC carriers. Such a change in release pattern may be due to the hydrophobic modification of CA by conjugation with PBC dye or potential ionic interaction between the dye and matrix functionalities [36]. A similar drug release pattern was observed for PCL-PEG and PCL-PEG-PBC formulations. The PCL-PEG released all of the drugs in a day while the release lasted up to 2 days for PCL-PEG-PBC. The hydrophobic modification to PCL-PEG by PBC conjugation slowed down the drug diffusion from the formulations. The PCL-PEG carriers were relatively more hydrophilic than CA carriers and showed variations in the release patterns of IND [37]. Figure 4d presents the 4-AP release pattern for different nanoparticles that followed a drug release pattern similar to IND. The 4-AP release from PCL-PEG and PCL-PEG-PBC was relatively quicker and lasted only for a day while it was sustained for CA and CA-PBC. The release of 4-AP from CA and CA-PBC carriers lasted up to 4 and 7 days, respectively. The carrier matrix hydrophilicity and drug nature dictated the timing and rate of 4-AP release. The solubility of IND in PBS (pH 7.4) is 0.05 mg/mL and a roughly five-fold increase was attained with CA-IND formulations with a value of 0.28 mg/mL. A 100 mg CA-IND formulation with 15.108 mg total IND content released the entire drug in 4 days due to increased solubility. However, freely water-soluble 4-AP formulations with identical drug content released the drug over 7 days. This enhanced IND solubility and 4-AP retention in the formulation was due to the novel properties of the dye-conjugated nanocarriers. Therefore, injectable nanocarrier formulations provide the ability to deliver the protic and aprotic drug molecules with different release profiles. Additionally, the nanoformulations improved the dissolution properties of poorly soluble drugs such as IND and provided sustained release of freely water-soluble drugs like 4-AP.

### 3.7. Drug Transport Studies

The fractional drug release (*M_t_*/*M_∞_*) up to 60% for both the drugs at time *t* were fitted with Higuchi and Korsmeyer–Peppas equations, and the values of *k* and *n* were calculated by the least-squares method (Table 2). The *n* values ranging from 0.5 to 1.0 are indicative of anomalous transport [29]. In this study, the *n* values ranged between 0.46 and 0.69, with correlation coefficients ranging from 0.93 to 0.99 indicating that the drug release deviated slightly from the Fickian transport [38]. The calculated values of k varied from 2.5 to 8.1, indicating potential drug–polymer interaction. In general, nanoparticles reach equilibrium water uptake in minutes, which converts the glassy polymer into a rubbery state. However, the drug release lasted for several hours, indicating that the polymer chain relaxation has less influence on the drug release, but is governed by molecular diffusion [23].

### 3.8. Nanoparticle Cell Uptake and Toxicity

Figure 5 presents the fluorescent images of the skin fibroblasts labeled with live–dead stains, nanoparticles, and layover cells on days 1 and 3 of in vitro culture. It is apparent from the images that fibroblasts maintained morphology with no evident cell death by the nanoparticle addition. Day 3 shows some cell death for CA-PBC-IND compared to 4-AP. The layover images show the presence of nanoparticles taken by the cells as well as in the culture media. Figure 6 presents the fibroblast metabolic rate over 3 days under the treatment of various nanoformulations. In general, cells treated with IND nanoformulations showed higher metabolic activity as compared to free IND treatments at equivalent doses (Figure 6a). However, with increasing culture time (by 72 h), a slight decrease in metabolic activity was observed for all groups including control drug treatments (Figure 6a). An identical response was also seen for the groups treated with 4-AP nanoformulations (Figure 6b). Collectively, these findings suggest nanoparticle uptake by fibroblasts and their viability over 3 days in culture, indicating the non-toxic nature of nanoparticles and their ability to be imaged in vitro.

### 3.9. In Vivo Nanoparticle Injection

To test the injectability of nanoparticles into negatively charged, dense tissues, nanoparticles were injected into coccygeal mouse IVDs in vivo. Nanoparticles were visualized using the DAPI filter. To identify the IVD tissue and cells, we used TdTomato mice. By slightly overexposing the Texas Red channel, we were able to image IVDs without additional staining. Confocal microscopy revealed that the nanoparticles remained within the extracellular matrix 1 day after injection (Figure 7A). Nanoparticles accumulated within the NP (yellow box) and were detected at the injection site of the annulus fibrosus (arrow). Some nanoparticle accumulation was observed at the vertebral endplates (asterisk) of the injected and control IVDs, which was an artifact that commonly occurred during IVD harvest and processing. Importantly, the notochordal cell band of the NP remained intact after nanoparticle injection and was comparable to PBS-injected IVDs (Figure 7A). These proof-of-concept data suggest that nanoparticles can be safely injected into mouse IVDs in vivo without causing structural damage.

### 3.10. In Vitro Nanoparticle Uptake by Degenerated Human NP Cells

Nanoparticle accumulation was observed in human degenerated NP cells after a 3-day co-culture (Figure 7B). Together with the in vivo data, these data suggest that a concentration of 0.015 μg/mL nanoparticle injection would be sufficient for localized drug treatment of IVDs as nanoparticles accumulated in degenerated, quiescent NP cells. However, long-term studies are needed to fully characterize the safety and efficacy of nanoparticle injection in human IVDs.

## 4. Discussion

The IVD is the largest avascular structure in the human body and its nutrient supply relies mainly on diffusion [39], making systemic delivery of therapeutic agents challenging. This study developed an injectable nanocarrier system for the local delivery of pharmacological agents [40,41]. We demonstrated that our nanoparticles were successfully injected into mouse IVDs in vivo, where they remained within the IVD tissue one day after surgery. After co-culture for up to 3 days, nanoparticles were taken up by cells and cell viability was maintained.

A wide range of nanocarriers and hydrogels have been explored to deliver pharmacological agents to IVD to promote healing or decelerate disease progression [42,43]. Here, we designed and tested the feasibility of two types of nanocarriers using CA and PCL-PEG as base polymers [44,45]. Both of these polymers are FDA-approved pharmaceutical excipients used in a variety of drug delivery formulations [46,47].

Injectable nanocarriers in the form of liposomes, micelles, and particulate systems are a popular choice in a variety of drug delivery applications. However, for IVD applications, injectable hydrogels are preferred to deliver pharmacological agents due to their relative ease of adaptation. These injected hydrogel drug carriers result in the burst release of the drug lasting 1 to 2 h and accumulate within IVD due to poor degradation and clearance features [48,49]. From the material perspective, hydrogels often occupy IVD space for a prolonged time, even after the encapsulated drug has been depleted. The remaining gel potentially prevents native matrix formation, and its degradation products could contribute to further matrix degradation [48,49]. This limits the repeated injection of hydrogels to continue the therapy [50]. Many injectable PLGA-based nanocarriers are attractive for the delivery of bioactive molecules but are known to degrade into acidic degradation byproducts that might cause adverse effects [23]. As an alternative, selected polymers in this study offered improved release profiles of the hydrophobic drug and retention of the drug for a longer duration for hydrophilic drugs [23,51]. Both polymers present functional groups on the backbone to chemically conjugate bioactive molecules as well as fluorescent molecules to enable imaging. The advantages of the current injectable nanocarriers not only allow sustained release of therapeutic agents, but over several days degrade faster and allow repeated injections to continue the treatment for IVD regeneration [52].

Many organic fluorescent molecules and inorganic molecules have been used frequently for imaging purposes [53]. Often due to photobleaching, many fluorophores lose their intensity, quickly resulting in a weaker signal [54]. Some fluorescent dyes and quantum dots, though effective, are not approved for clinical use [55]. An FDA-approved fluorescent dye is Indocyanine green. Indocyanine green is used to determine the cardiac output and liver blood flow [56]. However, this is a large molecule and presents challenges to conjugating it to a polymeric backbone due to the crowding and steric hindrance effects [57]. Here, we reported on the efficient method to synthesize citric acid-based PBC fluorescent small molecules and demonstrated the successful conjugation to the backbones of CA and PCL-PEG. The proposed structure for the PBC dye and its conjugation was confirmed by FTIR and proton NMR. The proposed cyclization reaction and formation of 6-membered rings are in agreement with published work [58]. Additionally, the presence of additional peaks for PBC dye, shifting of characteristic bands, and formation of new ester linkages confirm the conjugation of PBC dye on both CA and PCL-PEG backbones [59].

These dye-conjugated polymers were fabricated into nanoparticles by oil-in-water emulsion and a solvent evaporation technique [60,61]. Particles were spherical with rough surfaces, as reported earlier for both CA and PCL-PEG polymers [62]. The encapsulation efficiency of a drug as a nanoformulation is dependent on the nature of the drug during an emulsion fabrication process [26]. In general, aprotic drugs such as IND have low aqueous solubility and result in higher drug encapsulation as compared to water-soluble drugs. In our earlier report, we achieved an EE of 40 to 50% for IND in chitosan microparticles [29], which is lower than the 70 to 80% EE reported here (Figure 4). Chitosan microparticles were fabricated in water in oil emulsion followed by chemical cross-linking, as IND dissolves in the oil phase more than in the water phase [29]. It was also expected to have a lower EE for 4-AP due to its higher solubility in water as reported for the other water-soluble drugs [63].

Characteristics of the drug-loaded nanoparticle sizes, shapes, surface charges, and the nature of the drug itself, dictate formulation stability, particle uptake, and drug release patterns [64]. These spherical nanoparticles presented a hydrodynamic radius ~150–162 nm that is ideal for injection through small-gauge needles, such as the 33-gauge needle used for nanoparticle injection into IVDs in our in vivo experiments (Figure 7A), and for cellular uptake, for example, by native degenerated NP cells (Figure 7B), potentially via an endocytosis mechanism [65]. As such, the fabricated nanoparticles showed a very tight size distribution with reproducible drug release over a wide size distribution [66].

The zeta potential of the nanocarriers is an important parameter for their stability and influence on cell interaction and toxicity (Cell Prolif. 2015 Aug; 48(4): 465–474) (Arya SS et al., Front Bioeng Biotechnol. 2022 Apr 5;10:849464). The zeta potential for PBC-conjugated nanoparticles was negative and lower than the placebo polymeric nanoparticles in physiologic conditions [67]. These values are in agreement with previously reported injectable nanocarriers that ensure their stability in suspension and following injection in vivo [68]. Nanoparticles presenting negative zeta potential present an advantage in the initial adsorption onto the cell membrane and reduced cell toxicity, as reported earlier [69]. The molecular-level drug distribution within the nanocarriers ensured the amorphous nature of both drugs as determined by XRD diffractograms. The molecular-level drug distribution potentially interacts with the polymer matrix and was also evident with observed shifts in the characteristic FTIR and NMR peaks [70]. The reduced IND crystallinity in the nanoformulations overcomes its limitation on aqueous solubility and provides an effective solution to improve its bioavailability [71]. Solid-dispersion dosage forms are an effective way of improving hydrophobic drug-aqueous solubility [72].

The drug release from nanoformulations is dependent on the carrier matrix hydrophilicity, size, aspect ratio, charge, diffusion, and drug solubility. In hydrophilic polymers, the initial drug release is governed by the nanoparticle water uptake to reach equilibrium, then the later part of the release is solely governed by the diffusion process [73]. In general, nanoparticles reach equilibrium water uptake in minutes, which converts the glassy polymer into a rubbery state [74]. In contrast, for the tested drugs in this study, the release lasted for 4 days for IND and 7 days for 4-AP, indicating that the polymer chain relaxation has less influence on the drug release but is governed by molecular diffusion [75]. It is also important to note that drug release from PBC-conjugated polymeric nanoparticles was slower than placebo polymeric nanocarriers for the drugs. Such an observed effect was a hydrophobic modification of the matrix due to PBC conjugation, which is supported by the literature [60,76].

Nanoparticle uptake by cells is dominated by particle, size, shape, and zeta potential [77]. It is also important to assess cell viability following treatment with formulations to ensure cytocompatibility of the carriers and drug doses [78]. The common mode of nanoparticle uptake by cells is through endocytosis [41], although positive zeta potential leads to increased nanoparticle uptake due to the increased interaction with negatively charged cell membranes [79]. The negatively charged nanocarriers were internalized by the fibroblasts as evidenced by nanoparticle fluorescence. The live–dead assay-stained fibroblasts did not show any changes in regular cell morphology and were viable. Both CA and PCL-PEG polymers and their nanocarriers are reported to be cytocompatible, and often higher drug doses result in cytotoxicity and affect cell viability [80]. In the present study, we did not observe any changes in cell morphology and viability. This observation was also supported by MTS assay results, where the cell metabolic assay (proliferation) was comparable within different test groups at each time point and higher than the free drug controls of identical concentration. This indicated a potential free drug cytotoxic effect, while nanoformulations reduced such a cytotoxic effect. A slight decrease in metabolic activity was observed for nanoformulation-treated cells on day 3, indicating continued drug release and exposure. These findings are identical to earlier reports, where increasing nanoformulation exposure time decreased cell proliferation and viability [81]. A limitation of this study is the short duration of the in vivo mouse IVD injections. However, the motivation for this experiment was to investigate if the nanoparticles were injectable and would be incorporated into the ECM. Our data demonstrated successful nanoparticle injection into mouse IVDs, where the autofluorescence of the nanoparticle enabled the visualization of nanocarriers in the ECM. Moreover, their distinct fluorescence demonstrated accumulation in human NP cells. These findings are quite similar to other polymeric nanocarriers, dyes, metallic particles, and quantum dots [42]. An advantage of the injectable nanoformulations reported in this study is that they enable the delivery of pharmacological agents in very small volumes (1 μL) and are detectable based on their autofluorescence. However, the tested drugs were chosen based on their molecular properties, rather than basing them on their potential to treat degenerated IVDs. Future studies will focus on long-term studies using nanoparticles loaded with drugs to promote IVD health. The nanocarriers developed in this study provide a valuable tool to study the effect of pharmacological agents in small, compartmentalized tissues like mouse IVDs or tissues that require precise local injections. Additionally, the small volume injection of nanocarriers enables the injection of multiple doses to continue treatment, which is a major limitation of other injectable hydrogel-based systems.

## 5. Conclusions

We report on the efficient synthesis and characterization of PBC dye and PBC dye-conjugated CA and PCL-PEG nanoparticles for the delivery of a hydrophobic and a hydrophilic drug. The PBC conjugation increased matrix hydrophobicity, which resulted in a slower drug release rate for the hydrophilic (7 days) and a faster release rate for the hydrophobic (4 days) and the ability to detect them in tissues and cells. The structure of the nanoparticles allowed drug–polymer interaction and conversion into an amorphous state and its molecular level distribution within the nanoparticles. With a hydrodynamic radius of ~150 nm, the cytocompatible nanoparticles were injectable using a small 33-gauge needle which would allow injections into IVD tissue without causing structural damage. The majority of the formulations applied in IVD and joints in pre-clinical models are based on injectable hydrogels. Reported particulate delivery systems allow the release of both aqueous-soluble and -insoluble drugs for an extended time in tissue compartments without occupying the tissue volume and clearance. The ability to prolong the release of bioactive molecules and degradation in an IVD environment is currently not known for this system. Future studies will utilize specific drugs and track the fate of nanocarriers within the IVD. Hence fluorescent nanoformulations developed in this study may serve as a platform technology for the delivery of multiple therapeutic agents to compartmentalized tissue such as IVD.

## Figures and Tables

**Figure 1 jfb-14-00052-f001:**
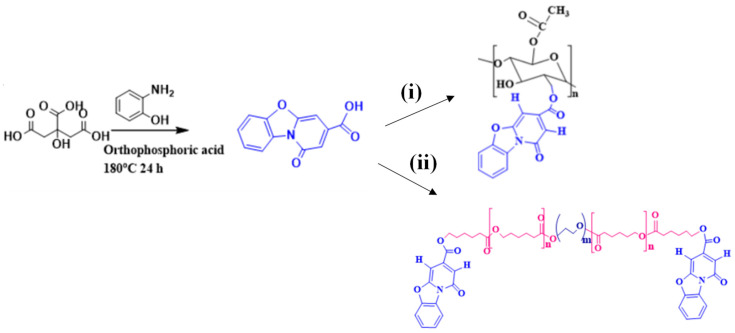
Synthesis of fluorescent dye PBC by the reaction of citric acid using precursor 2-Amino phenol. Molecular weights for PCL-PEG-PBC and CA-PBC are 11,849 and 55,853 respectively. Scheme (**i**): CA conjugated PBC with catalyst DCC and DMAP at 180 °C. Scheme (**ii**): PCL-PEG conjugated PBC with catalyst DCC and DMAP at 180 °C. DCC: N,N′-dicyclohexylcarbodiimide; DMAP: 4-dimethyl aminopyridine; DMF, dimethyl formamide. Nanoparticles of CA and PCL-PEG conjugated with a fluorescent dye were prepared by the oil-in-water emulsion technique to deliver high- and low-solubility 4-AP and IND drug molecules.

**Figure 2 jfb-14-00052-f002:**
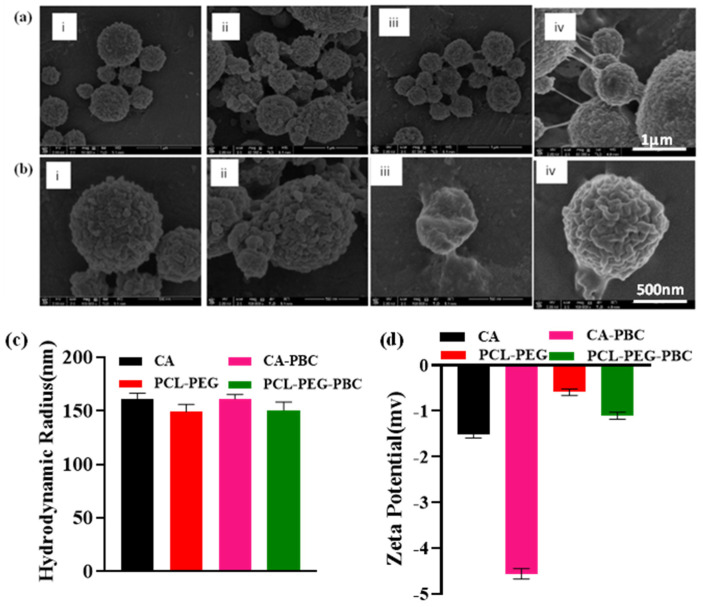
SEM images of fluorescent nanoparticles (**i**) CA, (**ii**) CA-PBC, (**iii**) PCL-PEG, and (**iv**) PCL-PEG-PBC at (**a**) 50,000× magnification (Scale bar 1 µm) and (**b**) 100,000× magnification (Scale bar 500 nm). All the nanoparticles show spherical shapes with an average particle size of ~150 nm. Particle sizes were comparable in both CA, CA-PBC, PCL-PEG, and PCL-PEG-PBC showing porous structure in all the nanoparticles. (**c**) Particle size analysis of CA, CA-PBC, PCL-PEG, and PCL-PEG-PBC (*n* = 5). (**d**) Zeta potential value of CA, CA-PBC, PCL-PEG, and PCL-PEG-PBC dispersed in PBS. Ordinary one-way ANOVA showed statistical difference with *p* value * = *p* < 0.0001 for both particle size and zeta potential for each group. The particle size DLS measurements showed the hydrodynamic radius varied between ~150–162 nm. The negative zeta potential of nanoparticles in PBS at pH 7.4 show their stability in suspension and advantage in cell diffusion.

**Figure 3 jfb-14-00052-f003:**
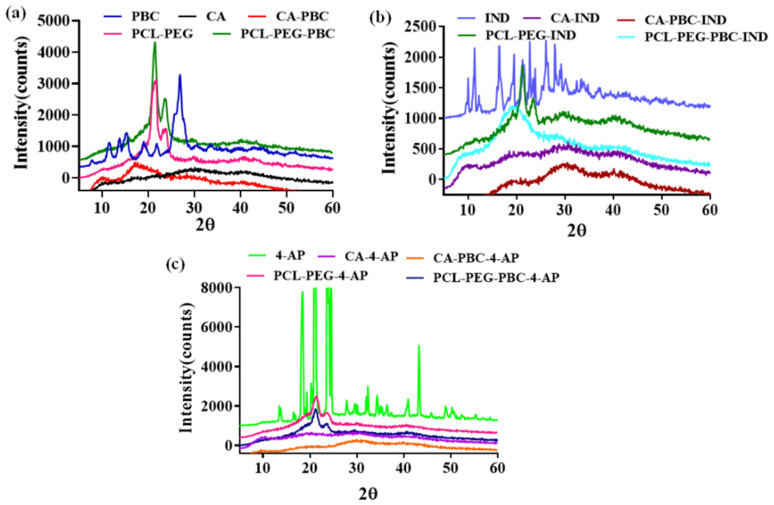
X-ray diffraction (XRD) spectra for (**a**) PBC, CA, CA-PBC, PCL-PEG, and PCL-PEG-PBC; (**b**) IND, CA-IND, CA-PBC-IND, PCL-PEG-IND, and PCL-PEG-PBC-IND; and (**c**) 4-AP, CA-4-AP, CA-PBC-4-AP, PCL-PEG-4-AP, and PCL-PEG-PBC-4AP. XRD was recorded from 5o to 60o (2θ) at a scanning speed of 0.2 deg/s. Cu-Ka radiation at 40 kV and 30 mA was used as the X-ray source. Nanoparticles show no sharp peaks indicating the amorphous nature, whereas the IND and 4-AP show sharp peaks indicating a crystalline nature, and drug-loaded nanoparticles show a diffused peak, indicative of the presence of drug matrix interaction.

**Figure 4 jfb-14-00052-f004:**
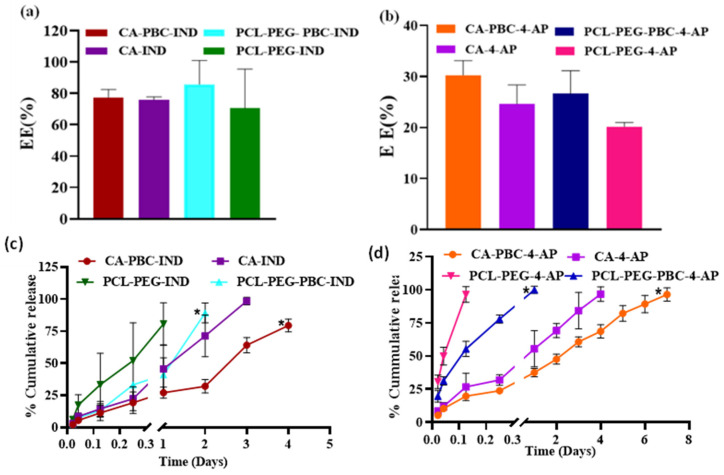
Encapsulation efficiency (EE (%)) of (**a**) IND and (**b**) 4-AP. (**c**) Cumulative release of IND from nanoparticles CA-PBC, CA, PCL-PEG-PBC, and PCL-PEG. The % EE of IND varied from 70 to 85% because of the hydrophobic nature of IND and drug interaction with the polymer carrier, while the % EE for 4-AP was very low and varied from 20 to 30% due to the hydrophilic nature of 4-AP. (**d**) Cumulative release of 4-AP from nanoparticles CA-PBC, CA, PCL-PEG-PBC, and PCL-PEG (*n* = 3 samples/group, mean ± s.d.). One-way ANOVA with mixed-effect analysis showed statistical difference with *p* value * = *p* < 0.0001 for CA-4AP vs CA-PBC-4-AP, PCL-PEG-4AP vs. PCL-PEG-PBC-4-AP, CA-IND vs CA-PBC-IND and PCL-PEG-IND vs. PCL-PEG-PBC-IND. The release of IND from these nanoparticles continued for up to 4 days, whereas the release of 4-AP lasting up to 7 days was dependent on the carrier and may be due to the hydrophobic modification of CA by conjugation with PBC dye or potential ionic interaction between the dye and matrix functionalities.

**Figure 5 jfb-14-00052-f005:**
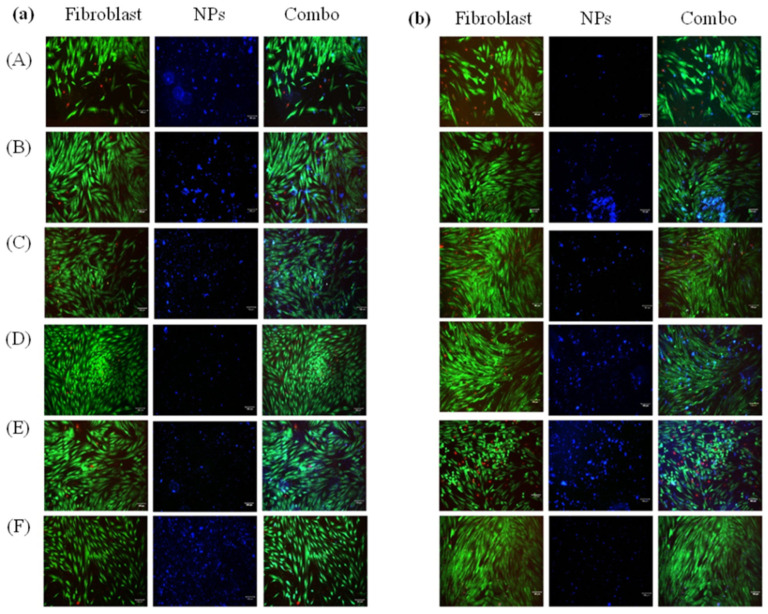
In vitro cell uptake of fluorescent nanoparticles of day 1 (**a**) and day 3 (**b**) with 4-AP (10 µg/mL) and IND (50 µg/mL) in drug media solution (*n* = 5). The left columns of each panel (**A**,**B**) show live/dead staining images of fibroblast cells. The middle column shows fluorescent nanoparticles (**A**) CA-PBC, (**B**) CA-PBC-IND, (**C**) CA-PBC-4-AP, (**D**) PCL-PEG-PBC, (**E**) PCL-PEG-PBC-IND, and (**F**) PCL-PEG-PBC-4-AP. The right columns for each panel (**a**,**b**) show combined (Combo) images. The layover images show the presence of nanoparticles taken by the cells as well as in the culture media. The fibroblast cells maintained morphology with no evident cell death by the nanoparticle addition from Day 1. Day 3 shows some cell death for CA-PBC-IND compared to CA-PBC-4-AP (scale bar 100 µm).

**Figure 6 jfb-14-00052-f006:**
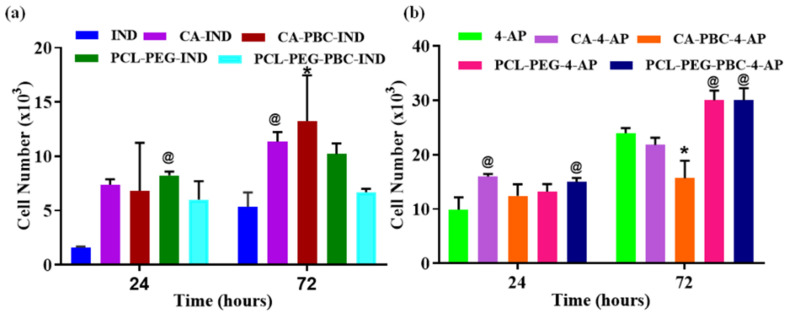
Cell proliferation (MTS assay) of fibroblast cells with drug alone (control groups: IND or 4-AP) or fluorescent nanoparticles with (**a**) IND (50 µg /mL), or (**b**) 4-AP (10 µg/mL) in drug media solution at 3 days. @ = *p* < 0.001 vs. respective control 4-AP, and * = *p* < 0.0001 vs. respective control *n* = 5. Two-way ANOVA with Tukey’s multiple comparisons showed: day 1 PCL-PEG-IND vs. IND (*p* < 0.001); day 3 CA-IND vs. IND (*p* < 0.001); CA-PBC-IND vs. IND (*p* < 0.001); day 1 CA-4-AP vs. 4-AP (*p* < 0.001); PCL-PEG-PBC-4-AP vs. 4-AP (*p* < 0.001); day 3 CA-PBC-4-AP vs. 4-AP (*p* < 0.0001); PCL-PEG-4-AP vs. 4AP (*p* < 0.001); and PCL-PEG-PBC-4-AP vs. 4-AP (*p* < 0.001). The nanoparticle uptake by fibroblasts and their viability over 3 days in culture indicate the non-toxic nature of the nanoparticles.

**Figure 7 jfb-14-00052-f007:**
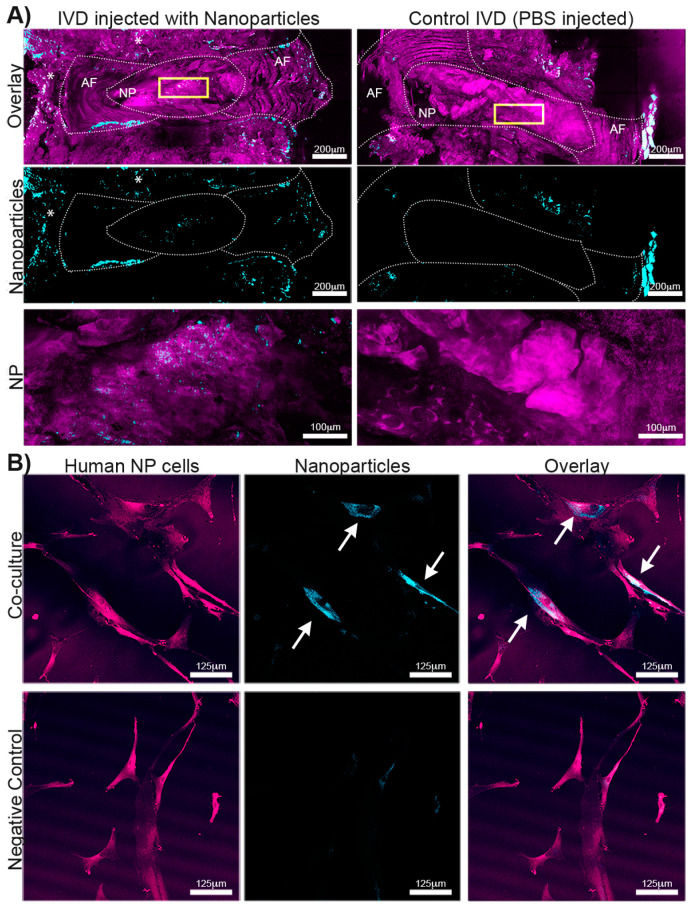
Nanoparticle incorporation in (**A**) extracellular matrices (ECM) of mouse intervertebral discs (IVD) in vivo and (**B**) degenerated human nucleus pulposus (NP) cells in vitro. (**A**) Nanoparticles (blue) were incorporated in the ECM of NP and annulus fibrosis (AF) of mouse IVDs (purple) 1 day after injection (left). Sample processing caused some artifacts in the surrounding tissues of the nanoparticle with PBS injection. Yellow boxes denote the region of interest (ROI) for NP nanoparticle incorporation. The arrow marks nanoparticle incorporation in the AF close to the injection site. Asterisks denote nanoparticle accumulation in the surrounding tissue (scale bar: 200 and 100 µm). (**B**) Nanoparticles (blue) accumulated in degenerated human NP cells (red) 3 days after co-culture. EP = Vertebral endplate. Nanoparticles were visualized using the DAPI channel; The TexasRed channel was used to visualize IVD tissue (TdTomato mice) and human NP cells (labeled with Cell Mask).

**Table 1 jfb-14-00052-t001:** FTIR spectra peaks with different functional groups for PBC, CA, CA-PBC, PCL-PEG, and PCL-PEG-PBC show the polymer and drug interaction. The CA, PCL-PEG FTIR spectra presented stretching vibrations for -C-O-C, -O-CH2-CH2, and -OH. The CA-PBC and PCL-PEG-PBC copolymer was confirmed by the formation of an ester bond between the PBC carboxylic group and the PCL-PEG hydroxyl group. The interactions between the functional groups for IND or 4-AP with polymeric nanoparticles were confirmed by the following peak interactions.

Functional Group	-C=O Str (cm^−1^)	-CH_2_ Str (cm^−1^)	NH_2_ Bend (cm^−1^)	NH_2_ Str (cm^−1^)	Amide (cm^−1^)	-OH (cm^−1^)
PBC	1722	2869	---	3138	1659	
CA	1738	2943	---	---	---	3479
CA-PBC	1746	2945	---		1668	3313
PCL-PEG	1733	2948	---	---	---	3538
PCL-PEG-PBC	1735	2946	---	---	1667	3322
IND	1713	2934	---	---	---	---
CA-IND	1748	2941	---	---	---	---
CA-PBC-IND	1756	2943	---	---	---	---
PCL-PEG-IND	1728	2947	---	---	---	---
PCL-PEG-PBC-IND	1724	2949	---	---	---	---
4-AP	---	---	1644	3444	---	---
CA-4-AP	1736	2941	1646	3404	---	---
CA-PBC-4-AP	1745	2943	1652	3445	---	---
PCL-PEG-4-AP	1732	2945	1649	3341	---	---
PCL-PEG-PBC-4-AP	1736	2949	1651	3345	---	---

**Table 2 jfb-14-00052-t002:** Drug release kinetics of IND and 4-AP from respective samples fit the Higuchi and Korsmeyer–Peppas drug release models. Higuchi models for both drug loading formulations show r^2^ values near unity indicative of a strong correlation to diffusion-controlled release mechanisms. Similarly, the *n* values ranged between 0.46 and 0.69, with correlation coefficients ranging from 0.93 to 0.99 indicating that the drug release deviated slightly from the Fickian transport.

Nanoparticles	Higuchi Model		Kinetic Model (Korsmeyer–Peppas)	
	k	r^2^	*n*	r^2^
CA-PBC-IND	4.0531	0.9133	0.5683	0.9606
CA- IND	2.5749	0.9689	0.6023	0.9538
PCL-PEG-PBC-IND	5.1732	0.9877	0.6823	0.9732
PCL-PEG-IND	5.8024	0.9951	0.6893	0.9743
CA-PBC-4-AP	3.9342	0.9908	0.4741	0.9867
CA-4-AP	2.490	0.9949	0.4646	0.9929
PCL-PEG-PBC-4-AP	8.062	0.9233	0.4804	0.9372
PCL-PEG-4-AP	6.0567	0.9699	0.6988	0.9983

**Higuchi model:** $\frac{M_{t}}{M_{0}}=Kt^{1/2}$, **Korsmeyer**–**Peppas:** $\frac{M_{t}}{M_{\infty }}=kt^{n}.$

## Data Availability

Data will be made available on request.

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
