# Peer review of "Novel Injectable Fluorescent Polymeric Nanocarriers for Intervertebral Disc Application"

_jfb, 2023, doi:10.3390/jfb14020052_

Round 1

Reviewer 1 Report

The manuscript entitled, “Novel Polymeric Injectable Fluorescent Drug Delivery Nanoparticles: A Proof-of-concept Study for Intervertebral Disc Application is a s proof-of-concept study that can benefit the researchers working in the area of intervertebral disc related diseases. However, the manuscript needs some work before being consided for publication in JFB. Therefore, major revision is recommended concerning the quality of draft.

Personally, I enjoyed reading the manuscript.

The suggestions are as follows:

The title is wordy and confusing, consider changing it to ‘Novel Injectable Fluorescent Polymeric Nanocarriers for Intervertebral Disc Application’

Abstract

Line 21: Add ‘,’ after ‘carboxylic acid (PBC)’

Line 23: Mention size based on SEM

Line 25: Mention zeta potential

Line 26: release ‘lasted’ or ‘lasting’

Keywords

Add two more key words

Introduction

Include market size of IVD medications

Line 56: Delete ‘an increase in’

Mention about how nanotech/nanocarriers have helped to shape the field of medicine.

Materials and methods

Line 123: Ratio of methanol/hexane solvent mixture?

Line 125: The sentence seems incomplete.

Line 127: Use unit for millimoles, mM

Results

Line 342: Somethings missing in the sentence or appears incomplete.

Line 343-345: Needs reference, consider https://doi.org/10.1016/j.ijbiomac.2022.06.145 where the shift in the stretching vibrations of -C=O in loaded-nanocarrier is evident compared to drug and empty carrier.

Figure 2: How is it possible that the hydrodynamic radius of the NPs is less than the SEM size? Form the SEM images I can say that the NPs are roughly > 500nm. The NPs for electron microscopy are desiccated hence should have even lesser size than that shown in hydrodynamic radius 150-162 nm. Please justify in the manuscript!

Line 387: What is mean by higher zeta potential? Please specify the value.

Line 397: What is meant by neat? Do you mean empty/unloaded? Better to use scientific terms.

Figure 4: Statistics not applied to EE%?

Figure 4d: % cumulative release is partially erased.

Discussion

Line 634-635: Needs reference, consider https://doi.org/10.3389/fbioe.2022.849464 which showcases that zeta potential is an important parameter for stability and toxicity.

Line 636: Replace neat with ‘empty/unloaded’ through the manuscript

Line 636: But how much negative? Also, stability of NPs is high if zeta potential >+30 or in your case <-30.

Line 660: Also, it depends on the aspect ratio and charge of NPs.

Please comment on the fate of carrier system? Are they degraded over time?

 Conclusion

First statement should be rephrased, particularly the beginning is grammatically incorrect.

Authors should also identify and mention the limitations of their current carrier system, giving a future direction to the research.

Reviewer 2 Report

 In this study, Some fluorescent polymeric nanoparticles were designed and synthesized, which can be served as a platform technology to deliver therapeutic agents to IVDs. There are still some points to be explained.

(1)   What is the molecular weight of PCL-PEG-PBC?

(2) In 2.5. Fabrication of nanoparticles and drug-loading, AP nanoparticles were prepared by the oil-in-water emulsion method? AP is a water-soluble compound, is this method suitable? I think oil-in-water emulsion method is suitable for the insoluble drug such as IND, but might not be suitable for the soluble drug? I think you should try other methods to improve the Drug Encapsulation Efficiency of AP.

(3)In 3.8. Nanoparticle Cell Uptake and Toxicity, it appears that the uptake of nanoparticles are very limited. And there is no explanation of uptake. Which polymer has best uptake rate? Besides, I did not find the results of  Nanoparticle uptake in Human NP cells, but there is a method section of 2.13. Nanoparticle uptake in Human NP cells.

 (4) In 3.9. In vivo Nanoparticle Injection, what is the nanoparticle group? CA-PBC or PCL-PEG-PBC? Why is there only one polymer group? And why is the blue color (Dapi for detection of PBC) different from the picture in cell uptake?Can you explain it ?

Round 2

Reviewer 1 Report

Authors have adequately incorporated the suggested changes. However, a couple of references cited in the text are not included in the reference list. I hope authors would rectify this mistake in the proofing stage.

Reviewer 2 Report

This manuscript can be accepted in this version.